# Microstructural Evolution of a Re-Containing 10% Cr-3Co-3W Steel during Creep at Elevated Temperature

Alexandra Fedoseeva [1,2,*], Ivan Brazhnikov [1], Svetlana Degtyareva [1], Ivan Nikitin [1] and Rustam Kaibyshev [2]

1. Laboratory of Mechanical Properties of Nanostructured Materials and Superalloys, Belgorod National Research University, Pobeda 85, 308015 Belgorod, Russia; 1216318@bsu.edu.ru (I.B.); 1390216@bsu.edu.ru (S.D.); nikitin_i@bsu.edu.ru (I.N.)
2. Laboratory of Prospective Steels for Agricultural Machinery, Russian State Agrarian University—Moscow Timiryazev Agricultural Academy, Timiryazevskaya, 49, 127550 Moscow, Russia; rustam_kaibyshev@bsu.edu.ru
* Correspondence: fedoseeva@bsu.edu.ru; Tel.: +7-4722-58-54-57

**Abstract:** Ten percent Cr steels are considered to be prospective materials for the production of pipes, tubes, and blades in coal-fired power plants, which are able to operate within ultra-supercritical steam parameters. The microstructural evolution of a Re-containing 10% Cr-3Co-3W steel with low N and high B content during creep was investigated at different strains at 923 K and under an applied stress of 120 MPa using TEM and EBSD analyses. The studied steel had been previously normalized at 1323 K and tempered at 1043 K for 3 h. In the initial state, the tempered martensite lath structure with high dislocation density was stabilized by $M_{23}C_6$ carbides, NbX carbonitrides, and $M_6C$ carbides. At the end of the primary creep stage, the main microstructural change was found to be the precipitation of the fine Laves phase particles along the boundaries of the prior austenite grains, packets, blocks, and martensitic laths. The remarkable microstructural degradation processes, such as the significant growth of martensitic laths, the reduction in dislocation density within the lath interiors, and the growth of the grain boundary Laves phase particles, occurred during the steady-state and tertiary creep stages. Moreover, during the steady-state creep stage, the precipitation of the V-rich phase was revealed. Softening was in accordance with the dramatic reduction in hardness during the transition from the primary creep stage to the steady-state creep stage. The reasons for the softening were considered to be due to the change in the strengthening mechanisms and the interactions of the grain boundary $M_{23}C_6$ carbides and Laves phase with the low-angle boundaries of the martensitic laths and free dislocations.

**Keywords:** high-chromium steels; creep; microstructural evolution; softening; strengthening mechanism; Zener pressure

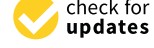



## 1. Introduction

Nine to twelve percent Cr martensitic steels are considered to be prospective materials for the production of pipes, tubes, blades, and other elements in coal-fired power plants, which are able to operate within ultra-supercritical steam parameters (873–893 K, 20–25 MPa) because of their excellent mechanical properties, low coefficient of linear thermal expansion, and reasonable prices [1–3]. The high level of creep resistance of these steels is attained by the formation of the tempered martensite lath structure (TMLS), which consists of prior austenite grains (PAGs), packets, and blocks with high-angle boundaries (HABs) and martensitic laths with low-angle boundaries (LABs), as well as the high dislocation density within the lath interior [4]. The stability of the TMLS in the initial state is reached by the precipitation of $M_{23}C_6$ carbides along all HABs and LABs of the TMLS, as well as MX carbonitrides (where M means V and/or Nb or their combination, X means C and/or N or their combination) randomly distributed in the ferritic matrix [5–10]. During creep at elevated temperatures, the precipitation of the fine Laves phase particles along

both HABs and LABs also stabilizes the TMLS [11–15]. However, the fast growth of the Laves phase can cause the formation of cracks [16].

The increase in the exploitation temperature of these steels is limited by high-temperature creep, which causes the recovery and recrystallization of the structure [1]. The appearance of the creep strength breakdown as the declination of the curve obtained using short-term creep data under high applied stresses leads to an overestimation of the 1% creep limit and 100,000 h creep strength [1–3]. Many researchers have related the appearance of this breakdown to: (i) the change in the creep/fracture mechanism [17–20], (ii) the replacement of MX carbonitrides by a large Z-phase [21–24], (iii) the change in the dispersion of $M_{23}C_6$ carbides [25,26], etc. Notably, changes in alloying, such as decreasing the N content to prevent Z-phase formation, increasing the W content to stabilize the MX, $M_{23}C_6$, and Laves phase, the addition of Re to increase W solubility in the ferrite, and/or increasing the B content to provide the improved dimensional stability of $M_{23}C_6$ carbides, do not overcome the problem of the creep strength breakdown [27–32]. We established the correlation between the creep strength breakdown and Laves phase coarsening using five experimental ingots [27,31]. However, for the Re-containing 10% Cr steel with low N and high B content, this correlation remains unclear [27–29] because there was no remarkable increase in the mean size of the Laves phase. On the other hand, we also found [31] that the distributions of the Laves phase and $M_{23}C_6$ carbides along the LABs of the martensitic laths and their mutual evolution had a more significant effect on the creep behavior than mean particle size alone, without consideration regarding their location. Therefore, the aim of the present study was to establish the reasons for the microstructural softening of the Re-containing 10% Cr–3% Co–3% W steel during creep at 923 K under a low applied stress (after the appearance of the creep strength breakdown), as well as to reveal the possible origin of the creep strength breakdown via the Laves phase evolution.

## 2. Materials and Methods

The chemical composition of the Re-containing 10% Cr–3% Co–3% W steel of (in wt.%). Fe-0.11C-9.9Cr-3.2Co-0.1Mo-2.9W-0.2V-0.07Nb-0.008B-0.002N-0.03Si-0.1Mn-0.17Ni-0.2Re was determined using a FOUNDRY-MASTER UVR optical emission spectrometer (Oxford Instruments, Abingdon, UK) and a METEK-300/600 nitrogen analyzer (METEKPROM Ltd., Izhevsk, Russia). The steel was subjected to homogenization annealing at 1423 K for 16 h and was hot forged at the same temperature, with air cooling. Then, after normalizing at 1323 K for 1 h, with air cooling, the final tempering at 1043 K for 3 h was carried out. Creep tests were carried out at a temperature of 923 K and under applied initial stresses ranging from 100 to 200 MPa with a step of 20 MPa until rupture. More detail regarding the creep tests was presented in a previous study [29]. Under an applied stress of 120 MPa, three interrupted tests (to 1001 h, 1%; to 5035 h, 2%; and to 10,001 h, 2.2%) were carried out. Vickers hardness was measured using an AFFRI DM-8 automatic microhardness tester with a load of 0.3 kg (Scicron Technology Co., Ltd., Bangkok, Thailand).

The structural investigation was carried out using a JEOL–2100 transmission electron microscope (TEM) (JEOL Ltd., Tokyo, Japan) and Quanta 600FEG scanning electron microscope (SEM) (FEI, Hillsboro, OR, USA), equipped with a high-speed EDAX Velocity$^{TM}$ EBSD camera (Gatan, Inc., Pleasanton, CA, USA). EBSD images were collected from an area of $50 \times 50\ \mu m^2$ with a step of 0.1 μm. The specimens for TEM and SEM analyses were prepared by electropolishing (293 K, 23 V) using a Struers "TenuPol-5" machine (Struers Inc., Cleveland, OH, USA). Carbon replicas for TEM were prepared using a Q 150REQuorum vacuum deposition machine (Quorum Technologies, Laughton, UK). The methods for the determination of the dislocation density, lath width, as well as the type, size, and particle number density of the secondary phase particles, were presented in the previous investigations [27–32] in detail. The equilibrium volume fractions of secondary phase particles at 1043 and 923 K were predicted using Thermo-Calc software (Version 5.0.4 75, Thermo-Calc software AB, Stockholm, Sweden, 2010).

## 3. Results

### 3.1. Initial Structure

After heat treatment, the tempered martensite lath structure (TMLS) with a mean PAG size of about $60 \pm 5$ μm was revealed (Figure 1). The description of the TMLS in the Re-containing 10% Cr–3% Co–3% W steel is given in [28] in detail. Information regarding the investigated phases is represented in Table 1. Some structural parameters are summarized in Table 2. The TMLS had a mean width of the martensitic laths of $290 \pm 30$ nm and high dislocation density within the lath interiors of $(2.0 \pm 0.5) \times 10^{14}$ m$^{-2}$ (Figure 1a, Table 2). The TMLS was stabilized by $M_{23}C_6$ carbides (Figure 1b), NbX carbonitrides (Figure 1c), and $M_6C$ carbides (not shown here but described in [28] in detail).

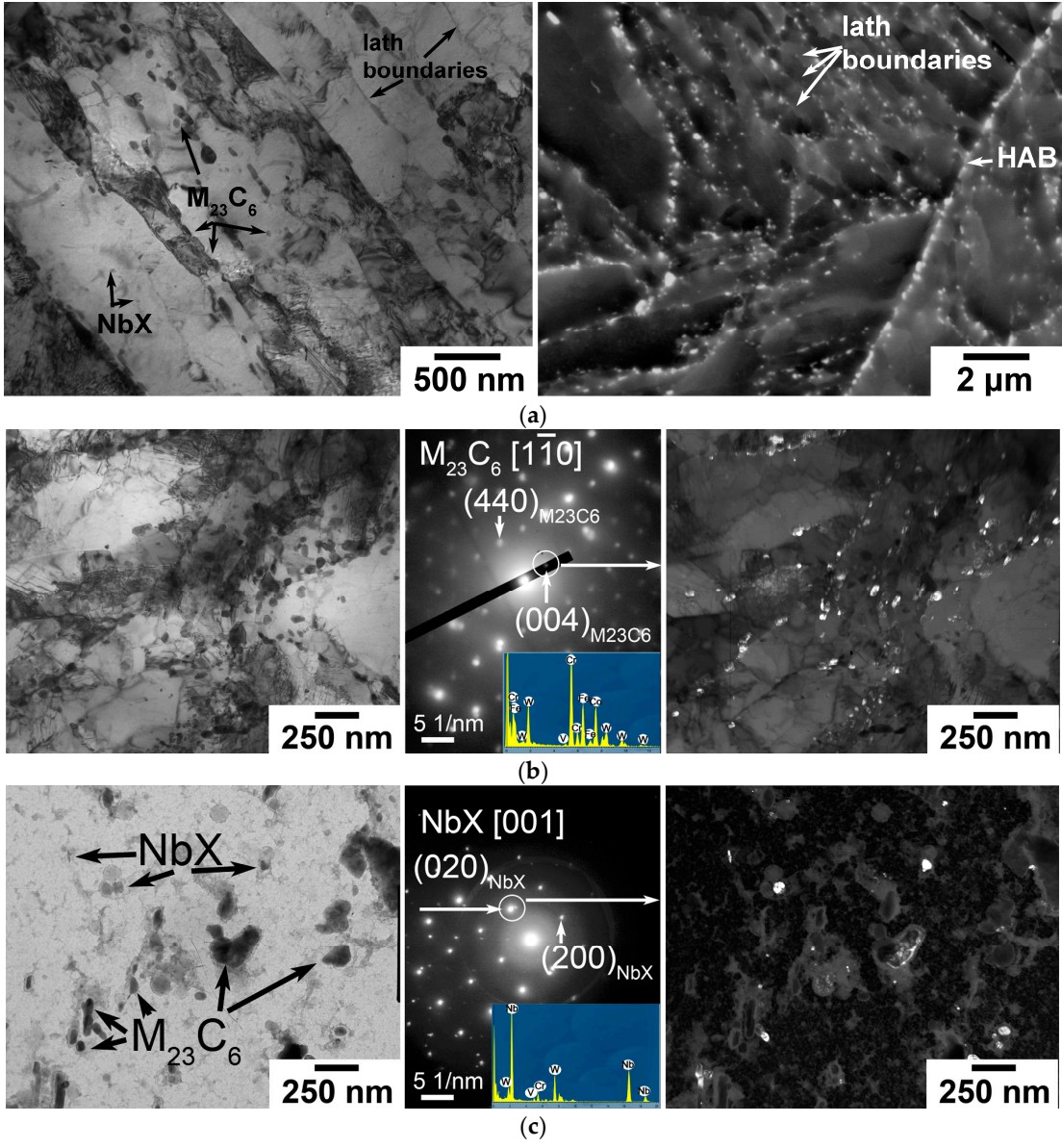

**Figure 1.** The TMLS of the Re-containing 10% Cr–3% Co–3% W steel: (**a**) TEM and SEM images; (**b**) the bright field image of $M_{23}C_6$ carbides decorating the lath boundaries together with electron diffraction pattern from the random $M_{23}C_6$ particle and dark field image in (004) $M_{23}C_6$ reflex obtained using TEM of foil; (**c**) the bright field image of $M_{23}C_6$ carbides and NbX carbonitrides together with electron diffraction pattern from the random NbX particle and dark field image in (020) NbX reflex obtained using TEM of carbon replica.

**Table 1.** A summary of the investigated phases in the work.

| Phase | Space Group | Bravias Lattice Type | Lattice Parameters, nm |
|---|---|---|---|
| $M_{23}C_6$ | $Fm\overline{3}m$ | Face-centered cubic | a = 1.0656 |
| NbC | $Fm\overline{3}m$ | Face-centered cubic | a = 0.44698 |
| $M_6C$ ($Fe_3W_3C$) | $Fd\overline{3}m$ | Face-centered cubic | a = 1.1087 |
| Laves ($Fe_2W$) | P6(3)/mmc | Hexagonal | a = 0.4727; c = 0.7704 |
| VN | Fm3m | Face-centered cubic | a = 0.413916 |

**Table 2.** The structural parameters of the studied steel after heat treatment and creep/aging conditions.

| Structural Parameters | | Tempered | Creep (1%; 1001 h) | Creep (2%; 5035 h) | Creep (2.2%; 10,001 h) | Creep (8.4%; 13,860 h) | Aged 13,860 h |
|---|---|---|---|---|---|---|---|
| Lath width, nm | | 290 ± 30 | 440 ± 30 | 660 ± 30 | 930 ± 30 | 950 ± 30 | 350 ± 30 |
| Subgrain size, nm | | – | 165 ± 30 | 430 ± 30 | 660 ± 30 | 760 ± 30 | – |
| Dislocation density within the lath interiors (TEM), $\times 10^{14}$ m$^{-2}$ | | 2.0 ± 0.5 | 1.8 ± 0.5 | 0.2 ± 0.05 | 0.2 ± 0.05 | 0.2 ± 0.05 | 1.6 ± 0.5 |
| Block size, μm | | 2.5 ± 0.3 | 3.1 ± 0.3 | 3.1 ± 0.3 | 2.9 ± 0.3 | 3.4 ± 0.3 | – |
| $S_v$, $\times 10^3$ m$^{-1}$ | | 524 | 528 | 508 | 1148 | 532 | – |
| $\theta$, deg | | 2.80 | 2.73 | 2.87 | 2.64 | 2.70 | – |
| Dislocation density inside the lath boundaries (EBSD), $\times 10^{14}$ m$^{-2}$ | | 1.5 ± 0.1 | 1.5 ± 0.1 | 1.5 ± 0.1 | 3.2 ± 0.1 | 1.5 ± 0.1 | – |
| Mean KAM value, deg | | 0.626 | 0.627 | 0.609 | 0.811 | 0.624 | – |
| Dislocation density via KAM, $\times 10^{14}$ m$^{-2}$ | | 8.8 ± 0.1 | 8.8 ± 0.1 | 8.6 ± 0.1 | 11.4 ± 0.1 | 8.8 ± 0.1 | – |
| $M_{23}C_6$ | Size (all), nm | 67 ± 10 | 70 ± 10 | 74 ± 10 | 83 ± 10 | 103 ± 10 | 80 ± 10 |
| | Size (LAB), nm | 69 ± 10 | 70 ± 10 | 81 ± 10 | 85 ± 10 | 85 ± 10 | 66 ± 10 |
| | Volume fraction along LABs, % | 1.5 ± 0.1 | 1.1 ± 0.1 | 0.6 ± 0.1 | 0.5 ± 0.1 | 0.4 ± 0.1 | 1.2 ± 0.1 |
| | Volume fraction *, % | 1.98 | | | 2.01 | | |
| Laves | Size (all), nm | – | 111 ± 10 | 173 ± 10 | 174 ± 10 | 212 ± 10 | 197 ± 10 |
| | Size (LAB), nm | – | 115 ± 10 | 160 ± 10 | 186 ± 10 | 236 ± 10 | 214 ± 10 |
| | Volume fraction along LABs, % | – | 1.0 ± 0.1 | 0.8 ± 0.1 | 0.7 ± 0.1 | 06 ± 0.1 | 1.1 ± 0.1 |
| | Volume fraction *, % | – | | | 1.85 | | |
| $M_6C$ | Size, nm | 28 ± 5 | | | – | | |
| NbX | Size, nm | 37 ± 10 | 34 ± 10 | 33 ± 10 | 30 ± 10 | 40 ± 10 | 40 ± 10 |
| | Volume fraction *, % | 0.077 | | | 0.068 | | |
| VX | Size, nm | – | – | 55 ± 10 | 72 ± 10 | 80 ± 10 | – |
| | Volume fraction *, % | – | – | | 0.0153 | | – |

* Equilibrium value estimated using Thermo-Calc software.

$M_{23}C_6$ carbide with a mean size of about 70 nm was found to be the dominant phase with the equilibrium volume fraction of 1.98% (Figure 1b, Table 2), which was located along all structural boundaries of the TMLS. The size of $M_{23}C_6$ carbides was independent of their location (along the LABs of the martensitic laths or the HABs of blocks, packets, and PAGs). $M_{23}C_6$ carbides were enriched by Cr, W, and Fe; their mean chemical composition was (in wt.%) 47% Cr, 23% Fe, and 30% W. In a previous study [28], we reported on the

Kurdjumov–Sachs orientation relationship between carbides and the ferritic matrix. Along the LABs of the martensitic laths, the particle number density of $M_{23}C_6$ carbides was $0.9 \pm 0.1 \ \mu m^{-1}$. The particle number density was estimated as the count of the particles per unit of the lath length projection [27]. Under the assumption that the martensitic laths form a three-dimensional cube with the grain edge length $D$, the length of the boundary per cell on the surface is $2D$, and the volume fraction of particles located along the LABs ($F_{LAB}$) can be estimated as follows [32]:

$$F_{LAB} = \pi \beta d^2 / 3D, \tag{1}$$

where $\beta$ is the number particle density (in $\mu m^{-1}$), $d$ is the mean particle size (in $\mu m$), and $D$ is the lath width (in $\mu m$). Using Equation (1), the volume fraction of $M_{23}C_6$ carbides located along the LABs of the martensitic laths comprised 1.5% that indicated 73% of all $M_{23}C_6$ particles were located along the LABs (Table 1).

The round-shaped NbX carbonitrides with a mean size of about 40 nm (Figure 1c, Table 2) and elongated W-rich $M_6C$ carbides with a mean size of about 30 nm (given in [28], Table 2) were also observed in the TMLS. Their volume fractions were negligible (Table 2).

The size of blocks was determined as the distance between the HABs using EBSD, and it was $2.5 \pm 0.3 \ \mu m$ (Figure 2, Table 2). According to [33], dislocation density inside the lath boundaries ($\rho^*$) could be estimated as follows:

$$\rho^* = 1.5 S_v \theta / b, \tag{2}$$

where $S_v$ is the lath area per unit volume (in $m^{-1}$), $\theta$ is the average lath misorientation (in rad), $b$ is Burger's vector (in m), suggesting that lath misorientations ranged from 2 to 5 deg [4] (Table 2). Dislocation density inside the lath boundaries was $1.5 \times 10^{14} \ m^{-2}$ that was similar to the value of density of free dislocations within the lath interiors estimated using TEM (Table 2). The lattice distortion was visualized via kernel average misorientation (KAM) (Figure 2). The average value of KAM was 0.626 deg, and dislocation density was $(8.8 \pm 0.1) \times 10^{14} \ m^{-2}$, estimated as follows [34]:

$$\rho_{KAM} = \frac{2\vartheta}{xb}, \tag{3}$$

where $\vartheta$ is the mean KAM value for the 1st neighbor (in rad), $x$ is the step of scanning (in m), and $b$ is Burger's vector (in m). Higher values of dislocation density estimated via KAM (Table 1) were caused by the fact that KAM included the geometrically necessary dislocations and low-angle lath boundaries with misorientations < 5 deg [34].

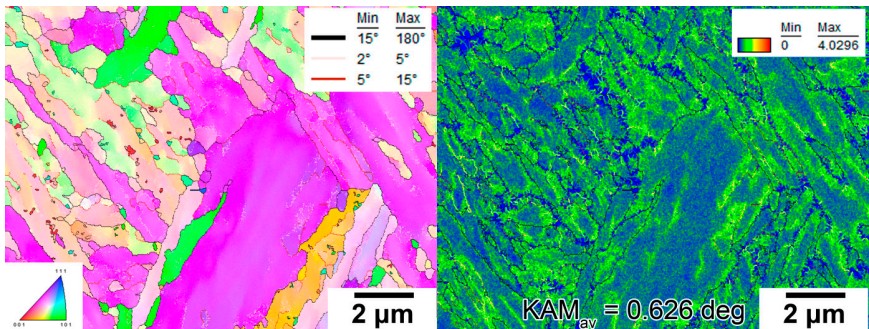

**Figure 2.** The EBSD images with KAM visualization of the Re-containing 10% Cr–3% Co–3% W steel in the initial state. The boundaries with misorientations from 2 to 5 deg are indicated by the light pink lines, from 5 to 15 deg by the dark red lines, and from 15 to 65 deg by the black lines. The data points with CI value < 0.1 were omitted.

### 3.2. Creep Properties and Hardness

The mechanical properties (creep and hardness) of the Re-containing 10% Cr-3%Co-3%W steel are represented in Figure 3. The appearance of the creep strength breakdown on the "applied stress vs. rupture time" curve (as the declination of the straight line obtained under the high applied stresses) was found to be a typical feature in 9–12% Cr martensitic steels [17–27]. For the Re-containing 10% Cr–3% Co–3% W steel, the creep strength breakdown was observed under the applied stresses below 140 MPa (Figure 3). For microstructural investigation in the region of low applied stresses, the creep test under 120 MPa was chosen. The creep curves under 120 MPa had three typical creep stages: (i) the primary creep stage from the start of the creep test to minimum creep rate $\dot{\varepsilon}_{min}$ corresponding to about 1000 h and ~1% strain; (ii) the steady-state creep stage from ~1000 h (1%) to ~5500 h (~2% strain); (iii) the tertiary creep stage from 5500 h (~2%) to the rupture corresponding to 13,860 h and 8.4% strain. The steady-state creep stage corresponded to a minimum creep rate of about $1.5 \times 10^{-10}$ s$^{-1}$. Three interrupted tests were carried out (Figure 3). The conditions of these tests are represented in Table 3. The first interrupted test was stopped when the primary creep stage was finished; minimum creep rate was attained (Figure 3, Table 3). The second interrupted test was stopped at the end of the steady-state creep stage; this state also corresponded to the minimum creep rate (Figure 3, Table 3). The third interrupted test was stopped during the tertiary creep stage (Figure 3, Table 3).

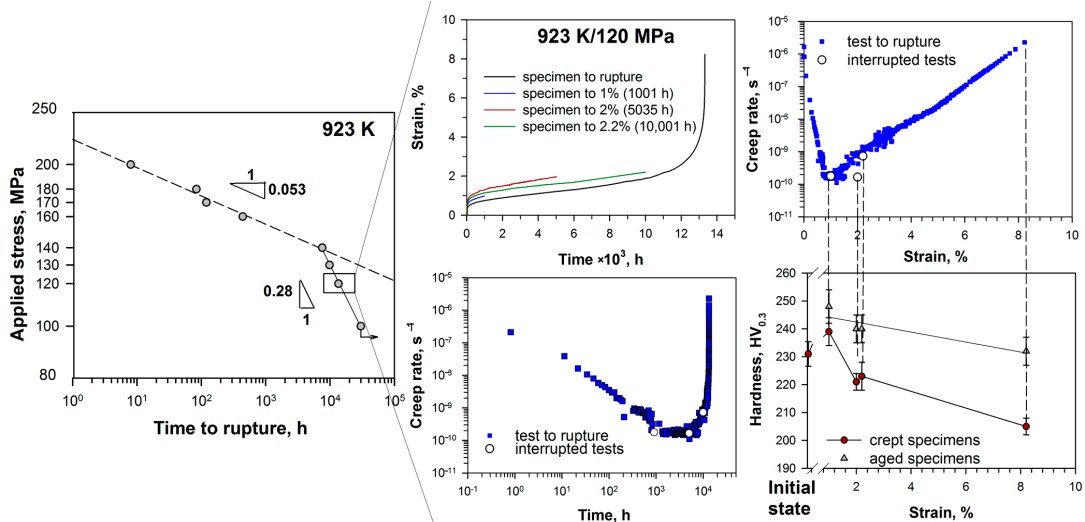

**Figure 3.** The creep properties of the Re-containing 10% Cr–3% Co–3% W steel together with the creep curves (strain vs. time, creep rate vs. strain, and creep rate vs. time) for the creep test under 923 K and 120 MPa to rupture with the points demonstrating interrupted tests to the strains of 1% (1001 h), 2% (5035 h), and 2.2% (10,001 h) and changes in hardness (HV$_{0.3}$) during the creep/aging tests.

**Table 3.** The conditions of the creep tests at 923 K and under the applied stress of 120 MPa.

| Conditions of the Creep Tests | 1 Interrupted Creep Test | 2 Interrupted Creep Test | 3 Interrupted Creep Test | Creep Test to Rupture |
|---|---|---|---|---|
| Time, h | 1001 | 5035 | 10,001 | 13,860 |
| Strain, % | 1.0 | 2.0 | 2.2 | 8.4 |
| $\dot{\varepsilon}$, s$^{-1}$ | $1.5 \times 10^{-10}$ | $1.5 \times 10^{-10}$ | $5.5 \times 10^{-10}$ | – |

The Vickers hardness test revealed two regions corresponding to (i) strengthening by 3.5% after the primary creep stage and then (ii) softening by 11% after the rupture in comparison with the tempered state (Figure 3). For aging for 1001 h, strengthening by 7%

was also revealed. At the same time, after aging for 13,860 h, no evidence for significant softening was observed compared to the initial state (Figure 3). So, creep remarkably reduced the strength of the material.

### 3.3. Microstructure after Aging (in the Grip Portions of Creep Specimens)

The TMLS with the mean lath width of 350 $\pm$ 30 nm was retained after aging for 13,860 h (Figure 4). Dislocation density within the lath interiors estimated by TEM was insignificantly reduced to $(1.6 \pm 0.5) \times 10^{14}$ m$^{-2}$ in comparison with the tempered state (Table 2). The Laves phase (Fe$_2$W) particles precipitated along the boundaries of the PAGs, packets, and blocks, as well as martensitic laths (Figure 4). The Laves phase was enriched by W, Fe, and Cr; the mean chemical composition was (in wt.%) 61% W, 31% Fe, and 8% Cr that corresponded to the equilibrium chemical composition of this phase predicted by Thermo-Calc software. This was accompanied by a decrease in the W content in the solid solution to about 1 wt.% [27]. The average size of the Laves phase particles was about 200 $\pm$ 10 nm regardless of their location (Table 1). Although the particle number density of the Laves phase located along the LABs of the martensitic laths was extremely low, at about 0.1 $\mu$m$^{-1}$, the volume fraction of this phase located along the LABs (Equation (1)) was about 60% of the equilibrium volume fraction (Table 1). M$_6$C carbides were not revealed in this state [27–29].

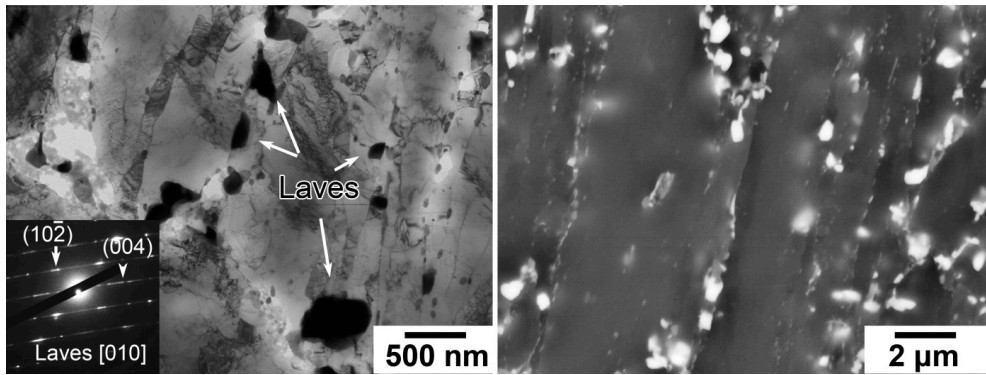

**Figure 4.** The microstructure (TEM and SEM images) of the Re-containing 10% Cr–3% Co–3% W steel after aging (in the grip portion of the creep ruptured specimen) at 923 K for 13,680 h.

The dispersions of M$_{23}$C$_6$ carbides and NbX carbonitrides were insignificantly changed (Table 2). Both volume fraction of M$_{23}$C$_6$ carbides located along the LABs and equilibrium volume fraction of NbX carbonitrides were slightly decreased (Table 2). These structural changes did not lead to softening of the studied steel during aging for 13,860 h compared to the tempered state (Figure 3).

### 3.4. Microstructural Evolution during Creep

- *Evolution of the dislocation and lath/subgrain structures*

During creep, the TMLS was strongly evolved (Figure 5, Table 2). Widening of the martensitic laths by 1.3 times due to the migration of Y-junctions [35] was revealed during the steady-state creep stage (Figure 5, Table 2). This was accompanied by the reduction in free dislocation densities within the lath interiors by one order of magnitude (Table 2). EBSD analysis revealed the significant increase in the block size from 2.5 $\pm$ 0.3 $\mu$m in the tempered state to 3.4 $\pm$ 0.3 $\mu$m in the creep ruptured state (Figure 6, Table 2). The mean KAM value was about 0.60...0.63 deg after tempering and creep conditions, except at 10,001 h (Figure 6, Table 2). The fraction of the LABs with misorientations ranging from 2 to 15 deg was insignificantly increased from 53% in the initial state to about 60% in the creep ruptured state that was accompanied by the decrease in the fraction of the HABs with misorientations > 15 deg (Figure 7). The average lath misorientation ($\theta$) remained at about 2.7 $\pm$ 0.2 deg from creep up to rupture (Figure 7, Table 2).

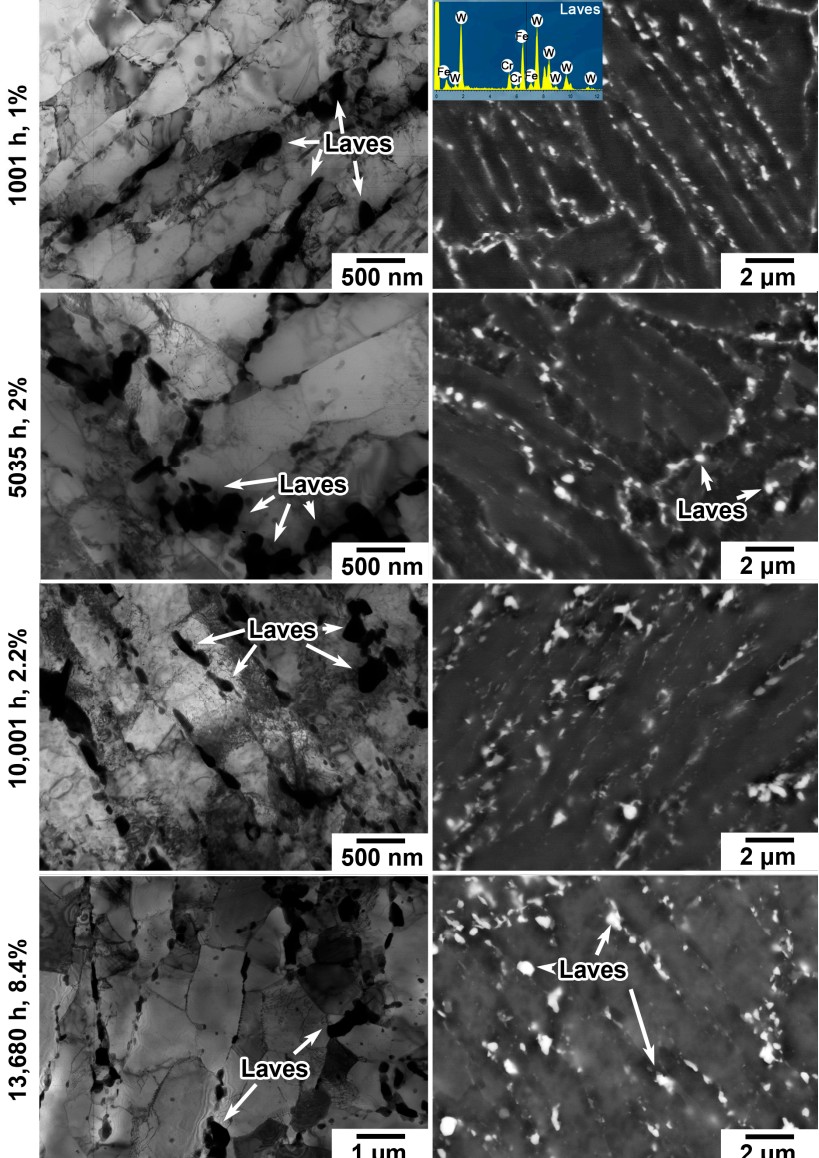

**Figure 5.** The microstructures (TEM and SEM images) of the Re-containing 10% Cr–3% Co–3% W steel after the creep tests to the strains of 1% (1001 h), 2% (5035 h), 2.2% (10,001 h), and 8.4% (13,680 h).

Notably, the increase in the fraction of the LABs with $\theta$ ranging from 2 to 5 deg had non-monotonic behavior: at the end of the primary creep stage, the fraction of these LABs increased by 12.9% compared to the tempered state; then, during the steady-state creep stage, their fraction was slightly decreased by 6.3%; further, during the transition from the steady-state creep stage to the tertiary creep stage, the remarkable increment in the fraction of such LABs by 52% was revealed; and finally, the fraction of such LABs was decreased again in the ruptured state (Figure 7). The reduction in dislocation density within the lath interior and increase in the fraction of the LABs were considered to be related processes because new low-angle boundaries have dislocation origins [36]. But the EBSD and TEM results were in contrast to each other: the strong reduction in dislocation density occurred before the significant increase in the fraction of the LABs with $\theta$ ranging from 2 to 5 deg (Figure 8). This is obviously caused by limits in EBSD analysis, which do not include the LABs with $\theta < 2$ deg. Moreover, in the tempered state, the laths with $\theta < 2$ deg were also present [37]. In our opinion, the evolution of both dislocations and LABs could be described as follows.

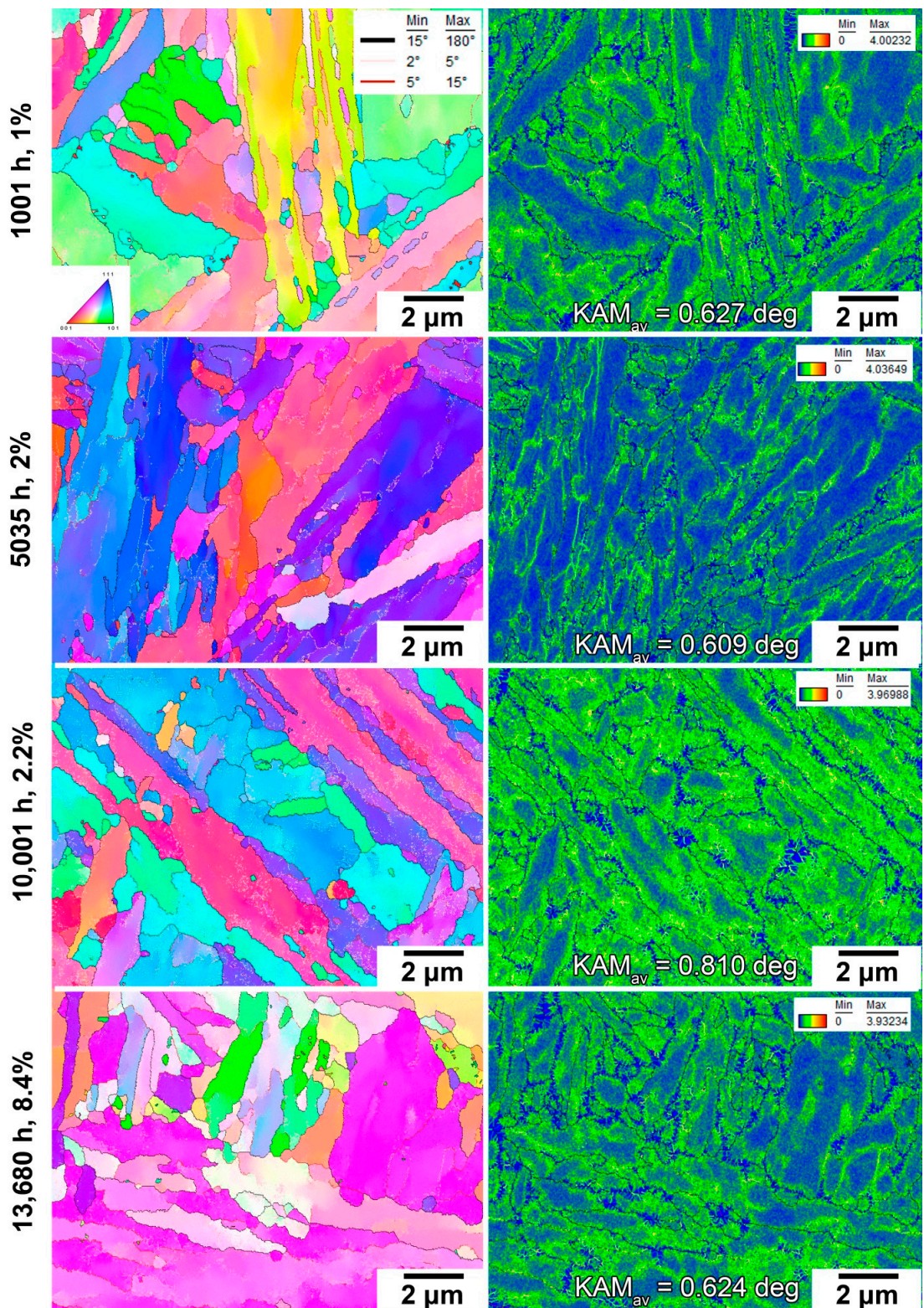

**Figure 6.** The EBSD images with KAM visualization of the Re-containing 10% Cr–3% Co–3% W steel after the creep tests to the strains of 1% (1001 h), 2% (5035 h), 2.2% (10,001 h), and 8.4% (13,680 h).

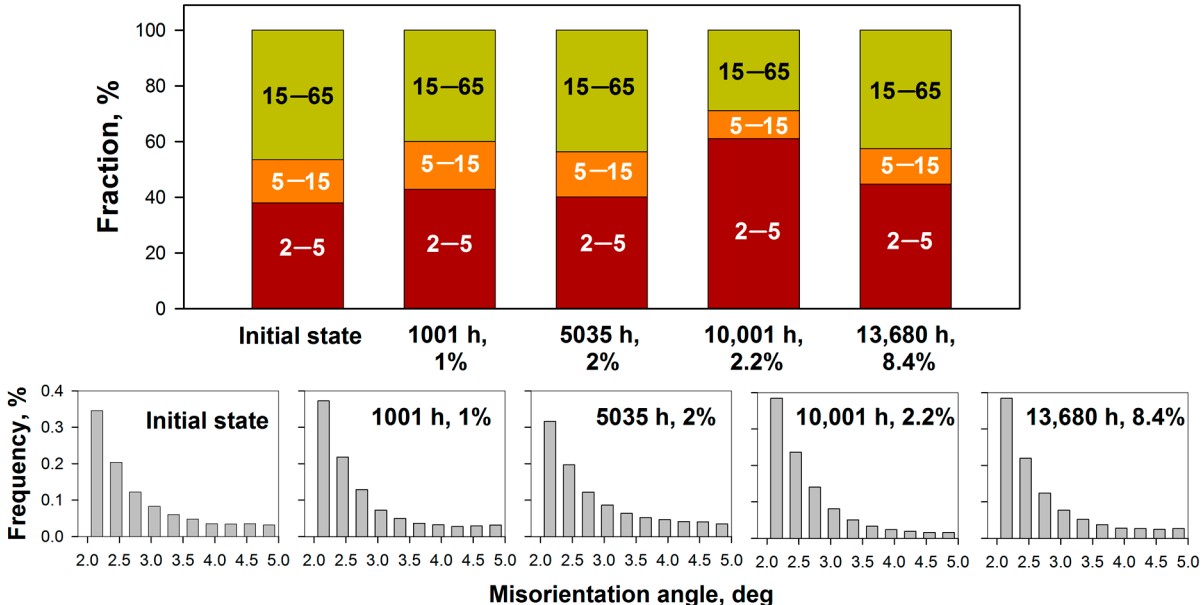

**Figure 7.** Changes in the ratio between the HABs and LABs together with distributions of misorientation angles ranging from 2 to 5 deg in the Re-containing 10% Cr–3% Co–3% W steel during creep test under 923 K/120 MPa.

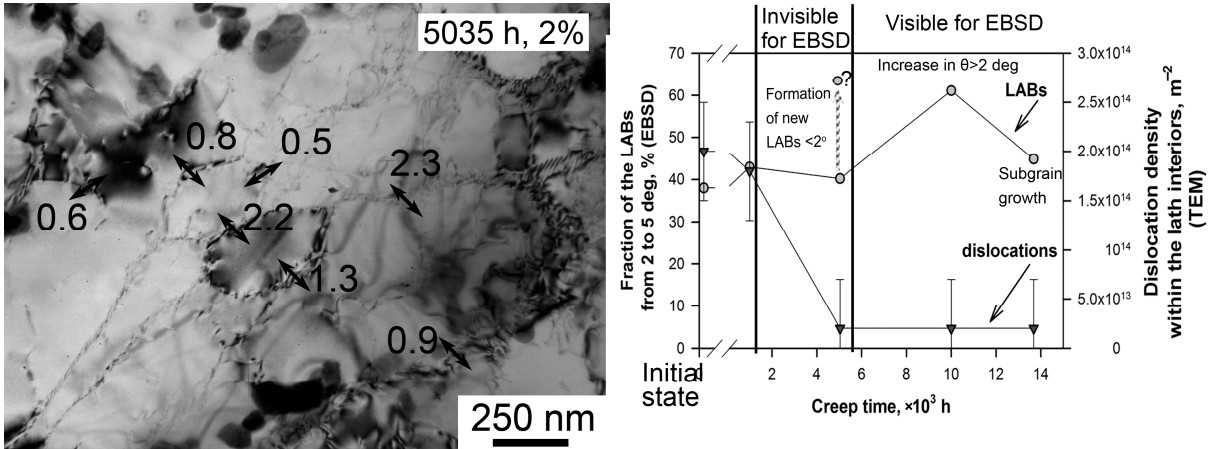

**Figure 8.** The formation of new LABs with $\theta$ < 2 deg at the steady-state creep stage (5035 h, 2.2%) together with changes in the LAB fractions with $\theta$ ranging from 2 to 5 deg estimated by EBSD and dislocation density within the lath interiors estimated by TEM.

At the primary creep stage, the reduction of dislocation density occurred due to the annihilation and rearrangement into dislocation walls [11,35,36]. The lath widening should lead to the decrease in the fraction of the LABs. However, this process was compensated by "old" tempered lath boundaries increasing their misorientations up to 2 deg and higher, and they became "visible" for EBSD analysis. As a result, the fraction of the LABs with $\theta$ ranging from 2 to 5 deg was increased. The formation of new "subgrains" with low misorientations < 2 deg remained unclear, since such "subgrains" were present in the structure, but we did not estimate their misorientations. "Subgrain" means an area which was surrounded by the LABs with $\theta$ < 5 deg and was located within the lath interior.

During the steady-state creep stage, the strong reduction in dislocation density by one order by magnitude was attributed to the formation of new subgrains with the low-angle boundaries with $\theta$ < 2 deg as is shown in Figure 8. These boundaries could be clearly distinguished in TEM, but for EBSD, they corresponded to an "invisible" region. EBSD analysis revealed the decrease in the LAB fraction that could be caused by the lath widening

to 660 ± 30 nm, boundaries which were taken into account by EBSD. In actual fact, the amount of LABs was significantly higher than EBSD can reveal; the fraction of the LABs corresponding to 5035 h of creep should be increased. During the transition from the steady-state creep stage to the tertiary creep stage (10,001 h, 2.2%), new boundaries with $\theta < 2$ deg increased their misorientation and became "visible" for EBSD analysis (Figure 8). Notably, the lath widening to almost 1 μm did not compensate the increase in the fraction of new subgrain boundaries with $\theta > 2$ deg (Table 2). The increment in the fraction of the LABs after 10,001 h of creep was apparent compared to the steady-state creep stage. Actually, the fraction of the LABs with $\theta$ ranging from 2 to 5 deg should correspond to the level of the steady-state creep stage or be slightly lower due to the lath widening. A further decrease in the LAB fraction was caused by the growth of subgrains only since the lath width retained a constant value (Figure 8, Table 2).

The estimation of dislocation density inside the lath boundaries via the lath area per unit volume and the average lath misorientation gave the same value of $(1.5 \pm 0.1) \times 10^{14}$ m$^{-2}$ except for 10,001 h of creep (Table 2). The value of dislocation density inside the lath boundaries for 5035 h of creep was found to be underestimated because of the formation of the additional subgrain area per unit volume, which was not taken into account by EBSD. Certainly, the average LAB misorientation should be decreased in this state. The values of dislocation density inside the lath/subgrain boundaries after 5035 h and 10,001 h of creep should be similar (Table 2).

In general, the comparison of TEM and EBSD results showed that EBSD analysis poorly described the materials with well-developed systems of low-angle boundaries with $\theta < 2$ deg.

- *Evolution of secondary phase particles*

During creep, the dispersions of the secondary phase particles were also evolved (Figures 9 and 10). The size of $M_{23}C_6$ carbides was dependent on their location (Figure 9, Table 2). Along the LABs, the size of $M_{23}C_6$ carbides monotonically increased from about 70 ± 10 in the tempered state to 85 ± 10 nm after rupture, whereas along both LABs and HABs, the size of carbides was remarkably increased during the transition from the steady-state creep stage to the tertiary creep stage (Figure 9). After rupture, the size of $M_{23}C_6$ carbides located along the LABs was lower by 17.5% than the size of these particles along both LABs and HABs. The particle number density of $M_{23}C_6$ carbides located along the LABs was significantly decreased after the end of the primary creep stage that decreased the volume fraction of this phase along the LABs by 2 times (Figure 9, Table 2). The evolution of size distributions of $M_{23}C_6$ carbides revealed that after 1001 h of creep, the fractions of the carbides with sizes less than 100 nm were 81% for the particles located along both LABs and HABs and 92% for the particles located along the LABs only (Figure 10). As creep time/strain increased, the fraction of the fine particles with sizes < 100 nm decreased to 58% for the particles located along both LABs and HABs and 73% for the particles located along the LABs only (Figure 10). This indicated that most fine particles were retained along the LABs up to rupture. Moreover, the fraction of the large carbides with sizes > 200 nm was increased from 0...1% to 8.4% for the particles located along both LABs and HABs and to 4.1% for the particles located along the LABs only as creep time/strain increased up to rupture (Figure 10). This meant that coarsening of this phase mainly occurred along the HABs.

The precipitation of the Laves phase along all types of TMLS boundaries was observed after all creep tests like aging. $M_6C$ carbides were not observed after all creep tests as was shown in [27–29]. In a previous study [27], it was reported that the W content in the solid solution reached the equilibrium value of about 1 wt.% after 500 h of creep. After 1001 h of creep, the Laves phase had the mean chemical composition of (in wt.%) 60% W, 23% Fe, and 17% Cr that was similar to the equilibrium chemical composition of this phase predicted by Thermo-Calc software. And further creep strain did not affect the chemical composition of this phase.

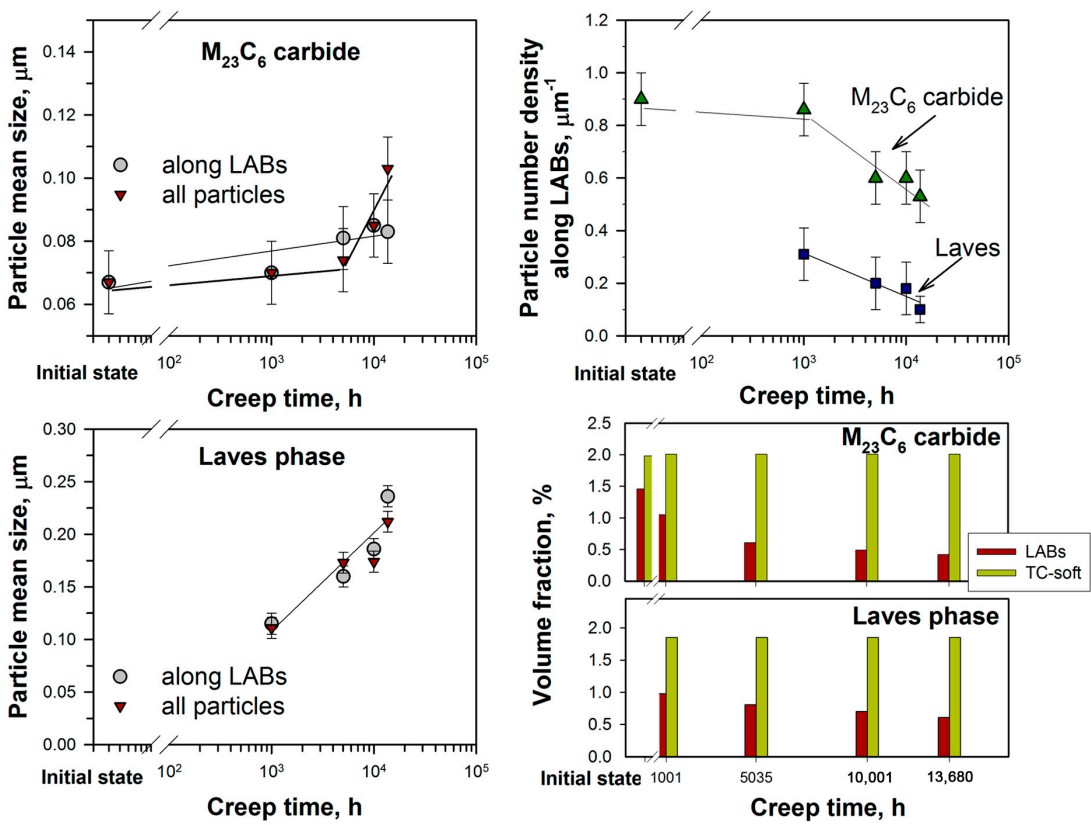

**Figure 9.** The evolution in the size and particle number density of the grain boundary $M_{23}C_6$ carbides and Laves phase together with the ratio of the volume fractions of the $M_{23}C_6$ carbides and Laves phase particles located along the LABs of the martensitic laths (indicated as LABs) to the equilibrium volume fractions estimated by Thermo-Calc software (indicated as TC-soft) in the Re-containing 10% Cr–3% Co–3% W steel during creep test under 923 K/120 MPa.

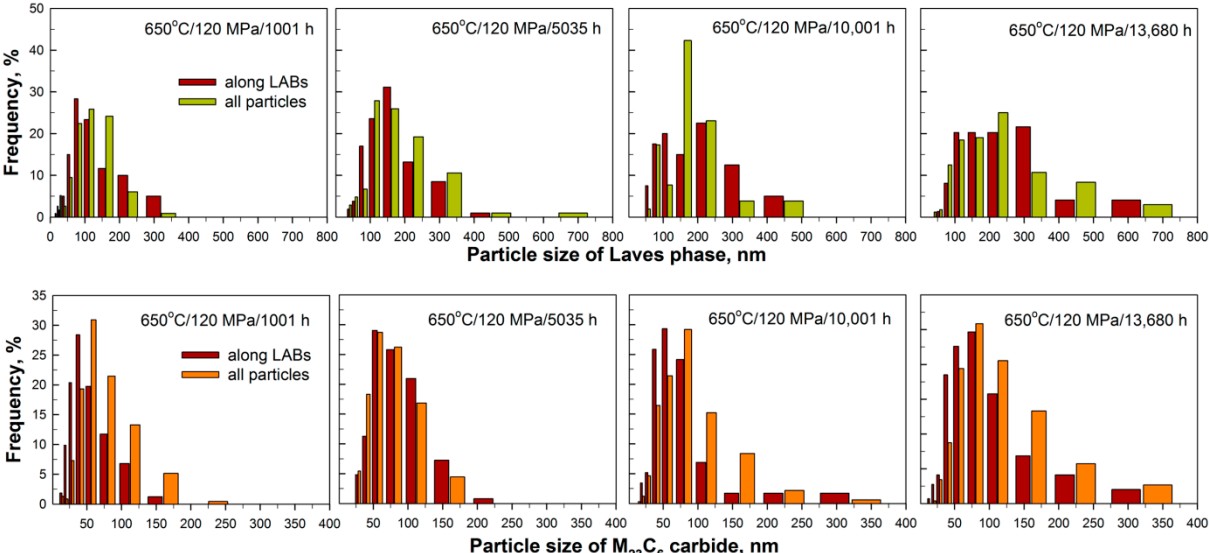

**Figure 10.** The evolution in size distributions of the Laves phase particles and $M_{23}C_6$ carbides in the Re-containing 10% Cr–3% Co–3% W steel during creep test under 923 K/120 MPa.

The size of the Laves phase was independent of their location: the mean size of this phase was the same along the LABs only as along both LABs and HABs. The growth of the Laves phase occurred monotonically from about $110 \pm 10$ nm after 1001 h of creep to about

$220 \pm 10$ after 13,860 h (Figure 9, Table 2). The particle number density along the LABs was strongly decreased at the steady-state creep stage that reduced the volume fraction of this phase along the LABs by 20% (Figure 9, Table 2). The evolution of size distributions revealed that after the end of the primary creep stage (1001 h, 1%), the fraction of the fine Laves phase particles along the LABs with sizes < 100 nm reached 50%, whereas at the end of the steady-state creep stage (5035, 2%), the fraction of the fine Laves phase particles along the LABs with sizes < 100 nm decreased to 23% (Figure 10). After rupture, only 10% of the Laves phase particles located along the LABs had sizes < 100 nm (Figure 10). Moreover, the fraction of the large Laves phase particles with sizes > 200 nm (regardless of their location) was increased from 10% after 1001 h to 50% after rupture (Figure 10). Coarsening of this phase occurred simultaneously along both LABs and HABs.

Therefore, during the steady-state creep stage, the dispersions of the $M_{23}C_6$ carbides and Laves phase located along the LABs were strongly degraded, increasing their mean size and decreasing the particle number density along the LABs.

The NbX carbonitride particles demonstrated high dimensional stability during creep, retaining their mean size of about $35 \pm 10$ nm (Table 2). The precipitation of a few separate V-rich MX carbonitrides was revealed at the end of the steady-state creep stage (5035 h, 2%) as is shown in Figure 11. These particles had an elongated shape and typical contrast (Figure 11a). The average chemical composition of the VX particles was (in wt.%) 44% V, 23% Nb, 23% Cr, and 10% Fe after 5035 h of creep. As the creep time/strain increased, the chemical composition of this phase remained unchanged (Figure 11b). The mean size of the VX phase increased from $55 \pm 10$ nm after 5035 h of creep to $80 \pm 10$ nm after 13,860 h. No evidence for Z-phase (CrVN) formation was revealed (Figure 11). The equilibrium volume fraction of this phase predicted by Thermo-Calc was 0.0153%.

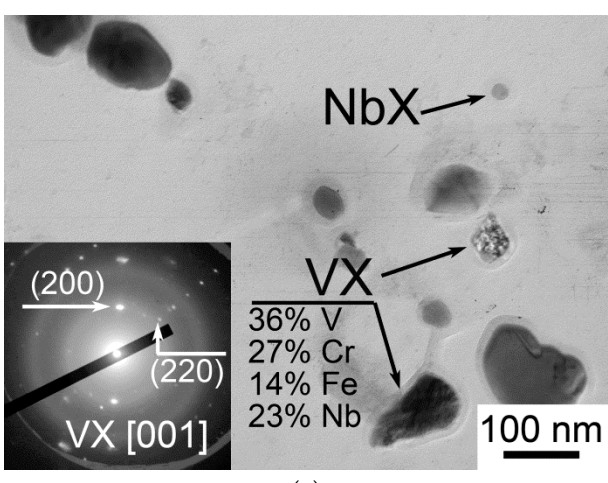
(**a**)

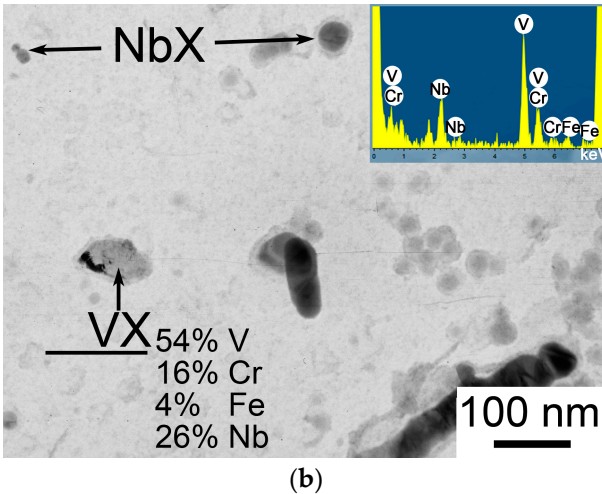
(**b**)

**Figure 11.** The precipitation of the VX phase in the Re-containing 10% Cr–3% Co–3% W steel: (**a**) at the end of the steady-state creep stage (2%, 5035 h); (**b**) at the tertiary creep stage (2.2%, 10,001 h).

## 4. Discussion

### 4.1. Softening of the Structure during Creep

Change in the hardness of the interrupted and ruptured creep specimens (Figure 3) revealed the increase in the hardness during the primary creep stage and the decrease in the hardness at the steady-state and tertiary creep stages up to rupture. Strengthening/softening can be estimated as follows:

$$\text{Strengthening/softening (in \%)} = (1 - H_i/H_o) \times 100\%, \tag{4}$$

where $H_i$ is Vickers hardness $HV_{0.3}$ after the interrupted and ruptured creep tests, and $H_o$ is Vickers hardness $HV_{0.3}$ in the initial state. Figure 12 demonstrates the temporal dependence

of softening/strengthening estimated via change in the $HV_{0.3}$ during the creep test under 923 K/120 MPa. There is a well-known relationship between the Vickers hardness and yield strength (YS), which can be written as follows [38]:

$$YS = 9.81HV/3.\tag{5}$$

The room yield strength of the Re-containing 10% Cr–3% Co–3% W steel is $560 \pm 50$ MPa [39], and the $HV_{0.3}$ is 231 (Figure 3). The relationship between the Vickers hardness and YS for the studied steel can be found as follows:

$$YS = 9.81HV/4,\tag{6}$$

which is similar to Equation (5). Using Equation (6), the Vickers hardness can be transformed into the YS after creep tests. This allows comparing the experimental YS with the strengthening mechanisms.

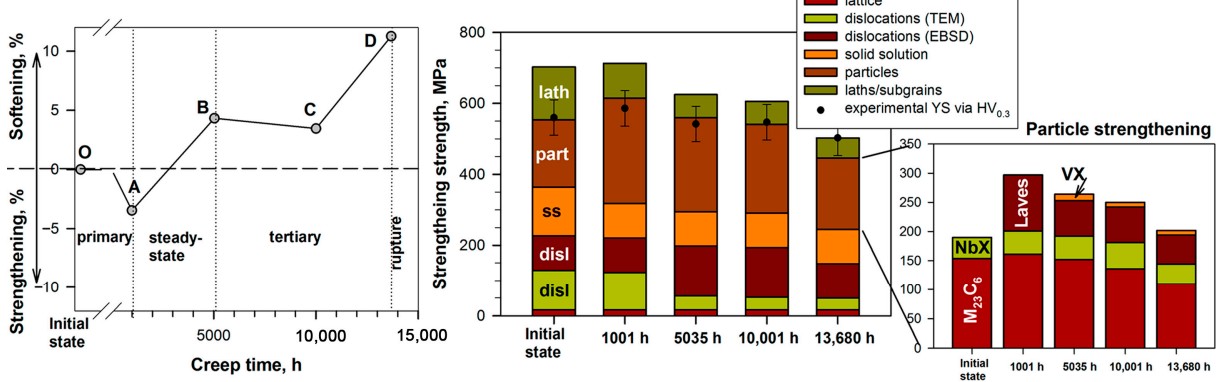

**Figure 12.** The temporal dependence of softening/strengthening estimated via change in the $HV_{0.3}$ during creep together with changes in the contributions to strengthening mechanisms.

The linear sum of strengthening mechanisms from lattice friction, dislocation density including dislocations within the lath interiors estimated by TEM and inside the lath boundaries estimated by EBSD, solid solutions due to Cr, W, Mo, and Co atoms in the ferrite, the secondary phase particles ($M_{23}C_6$, NbX, Laves phase, and VX), and laths/subgrains [40–45] shows good coincidence with the experimental YS (Figure 12).

The lattice friction is estimated as $2 \times 10^{-4} \times G$, where $G$ is a shear modulus (8.4 GPa at 293 K) [40], and it is about 17 MPa.

Dislocation strengthening is estimated suggesting the interactions of free dislocations with forest dislocations using Taylor theory as follows [41]:

$$\sigma_{disl} = \alpha Gb\sqrt{\rho + \rho^*},\tag{7}$$

where $\alpha$ is an iron polycrystalline constant (0.38), $\rho$ is dislocation density within the martensitic lath interiors, and $\rho^*$ is the dislocation density inside the lath boundaries ($m^{-2}$). After 5035 h and 10,001 h of creep, the value of dislocation density inside the lath boundaries of $3.2 \times 10^{14}$ $m^{-2}$ is used as discussed in Section 3.4.

Solid solution strengthening resulting from Cr, W, Mo, and Co atoms in the ferrite of the studied steel is evaluated as the linear sum as follows [42,43]:

$$\sigma_{ss} = \sum_{i=1}^{n} K_i C_i^{\frac{3}{4}},\tag{8}$$

where $K_i$ is a coefficient of solid solution strengthening: for Cr = 9.65 MPa/%$^n$, for W = 76 MPa/%$^n$, for Mo = 66 MPa/%$^n$, for Co = 2.0 MPa/%$^n$, $C_i$ is the concentration of elements in the ferrite (in atomic %).

Particle strengthening due to the $M_{23}C_6$ carbides and NbX carbonitrides in the initial state, as well as the $M_{23}C_6$ carbides, NbX, Laves phase, and VX carbonitrides in the creep states, as the linear sum is estimated in accordance with the Orowan model as follows [44]:

$$\sigma_{particle} = \frac{0.84Gb}{\sum_{i=1}^{n} d_i \sqrt{\frac{\pi}{6f_i}} - d_i}, \tag{9}$$

where $d_i$ and $f_i$ are the size and volume fraction, respectively, for the different types of the secondary phase particles. The volume fractions for the secondary phase particles are taken from Thermo-Calc software.

Lath/subgrain strengthening is evaluated using Langford–Cohen model as follows [45]:

$$\sigma_{lath} = 86.2 \times (2l)^{-1}, \tag{10}$$

where $l$ is the lath width (for the tempered state and creep states for 1001 h and 5035 h, as lath structure is preferred) or subgrain size (for creep states for 10,001 h and 13,680 h, as new subgrain structure occurs and the mean size of subgrains becomes smaller than the lath width).

The comparison of strengthening/softening of the Re-containing 10% Cr–3% Co–3% W steel during creep with change in the contributions of strengthening mechanisms to the experimental room YS revealed three regions as follows.

The first region OA corresponds to the primary creep stage (up to 1001 h of creep). The main structural changes in this region are found to be (i) the decrease in solid solution strengthening from 137 MPa in the tempered state to 97 MPa after 1001 h of creep (Figure 12) due to the decrease in the W content in the ferritic matrix from 3 wt% to about 1% [27], and (ii) the precipitation of the fine Laves phase particles that provides the increment in particle strengthening by 56% (Figure 12) in which case the dispersions of the $M_{23}C_6$ carbides and NbX remain unchanged (Table 2). The increase in particle strengthening compensates the decrease in solid solution strengthening that provides overall strengthening of the structure by 3.5% during the primary creep stage. A similar situation is observed for the aged states, where strengthening of the TMLS is also found after 1001 h of aging (Figure 3). Moreover, these mutual processes are considered to be main evolution changes during the primary creep stage in all 9–12% Cr steels that determine the slope of the primary creep stage [5,11,35].

The second region AB (Figure 12) corresponds to the steady-state creep stage. The main structural changes in this region can be described as follows.

(i)   The reduction in dislocation density within the lath interiors (Table 2) decreases their contribution to dislocation strengthening by 2.7 times. On the other hand, the dislocation density inside the lath/subgrain boundaries remains at a high level of $10^{14}$ m$^{-2}$ due to the formation of new subgrain boundaries with low misorientation (Figure 8).

(ii)  The growth of the martensitic lath width to $660 \pm 30$ nm decreases lath/subgrain strengthening by 34%. This is caused by the action of both applied stress and evolution of the grain boundary particles.

(iii) The increase in the mean size of the Laves phase particles decreases their contributions to particle strengthening by 36%.

(iv)  The formation of the VX phase slightly increases particle strengthening.

It should be noted that these main structural changes in the studied steel occur during the steady-state creep stage between 1001 and 5035 h of creep (Figures 5–10 and 12, Table 2). This is the principal difference from the microstructural evolution of the 9–12% Cr steels during creep under high applied stresses, where remarkable structural degradation is revealed during the tertiary creep stage [5,11,36,46]. The formation of a cellular dislocation network is also revealed in simple 0.2% C and 9% Cr steel during the primary creep stage, whereas their stability during the steady-state creep stage is considered to be key internal

back-stress against creep strain [36]. This is in accordance with the present observation in the studied Re-containing 10% Cr steel.

The third region BD corresponds to the tertiary creep stage (Figure 12). Notably, during the transition from the steady-state creep stage (5035 h) to the tertiary creep stage (10,001 h), no evidence for softening is revealed, while the lath width increases from $660 \pm 30$ to $930 \pm 30$ nm, respectively. It is possible that "new" subgrain boundaries play the same role as "old" lath boundaries, preventing the movement of free dislocations. Slight changes in the dispersion of the grain boundary particles do not cause softening of the structure (Figure 12). However, a dramatic reduction in the strength is found after rupture of the creep specimen (Figures 3 and 12). During the tertiary creep stage (from 10,001 h to rupture), the next microstructural changes are found as follows:

(i)  He growth of subgrains due to the applied stress ("new" subgrain boundaries are free from the secondary phase particles) causes the decrease in dislocation density inside the lath/subgrain boundaries due to the reduction in the lath area per unit volume when retaining the average lath/subgrain misorientation of 2.7 deg (Table 2). This provides the decrease in both dislocation strengthening by 29% and lath/subgrain strengthening by 14% (Figure 12).

(ii)  Coarsening of the grain boundary particles is considered to be the main reason for the growth of the lath width. The increase in the size of the $M_{23}C_6$ carbides and Laves phase by 18...19% provides the decrease in particle strengthening by 20% (Figure 12, Table 2).

In other 9–12% Cr steels, the mutual growth of the laths and grain boundary carbides and Laves phase also leads to the loss of stability of the TMLS during the tertiary creep stage up to the full transformation of the lath structure into a 100% polygonal subgrain structure [5,11,27,31,32,36,46,47].

### 4.2. The Interactions of Secondary Phase Particles with Lath Boundaries amd Dislocations

- *Zener retarding force from grain boundary particles*

The mutual growth of the lath width and grain boundary particles can be described in terms of the balance between driving force due to the migration of Y-junctions of the lath boundaries and Zener retarding forces for recrystallization processes [44]. For the $M_{23}C_6$ carbides and Laves phase located along the LABs of the martensitic laths, Zener retarding force ($P_{LAB}$) can be described as follows [32]:

$$P_{LAB} = \frac{\gamma F_{LAB} D}{d^2}, \tag{11}$$

where $\gamma$ is the surface energy per unit area of the lath/subgrain boundary, and it is 0.153 J m$^{-2}$, $F_{LAB}$ is the volume fraction of the particles located along the LABs estimated by Equation (1), $D$ is the lath width, $d$ is the mean size of the particles located along the LABs (Table 2).

For the MX (NbX + VX) carbonitrides, which are randomly distributed in the ferritic matrix, Zener retarding force is described as follows:

$$P_z = \frac{3\gamma F}{d}, \tag{12}$$

where $F$ is the volume fraction predicted by Thermo-Calc software, and $d$ is the mean particle size (Table 2). Figure 13a demonstrates changes in Zener retarding forces from the different kinds of secondary phase particles during creep.

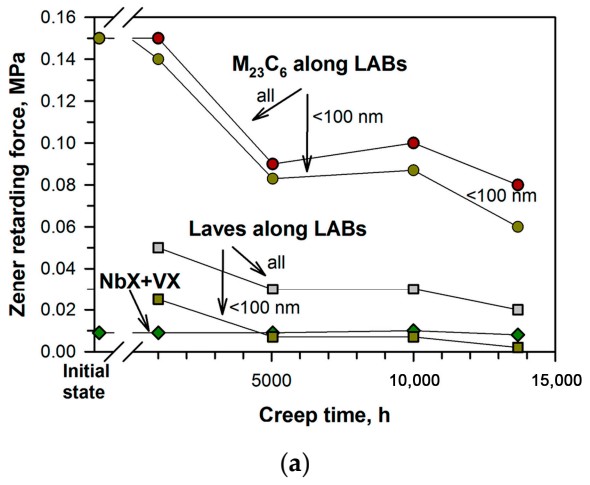
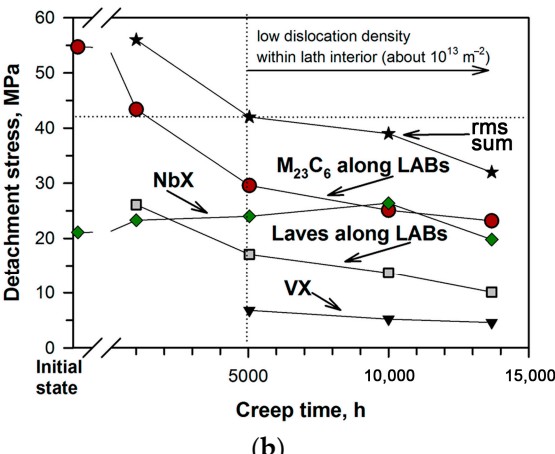

(**a**)  (**b**)

**Figure 13.** The interaction of the secondary phase particles with the lath boundaries and free dislocations: (**a**) change in Zener retarding forces; (**b**) change in the stress of detachment of the dislocation from the attractive particles after finishing the climb during creep under 923 K/120 MPa.

The remarkable reduction in Zener retarding forces from the grain boundary particles located along the LABs is revealed at the end of the steady-state creep stage (5035 h). The linear sum Zener retarding force caused by the $M_{23}C_6$ and Laves phase decreases from 0.15 MPa in the tempered state to 0.12 MPa after 5035 h, and then it slightly decreases to 0.10 MPa after rupture (Figure 13a). Notably, after the primary creep stage, the sum of the Zener forces due to the $M_{23}C_6$ and Laves phase located along the LABs increases to 0.20 MPa because of the precipitation of the fine Laves phase particles (Figure 13a). Zener force from the MX (NbX + VX) carbonitrides is very low and remains at about 0.01 MPa up to fracture (Figure 13a).

The balance between driving force and Zener retarding force provides the equilibrium lath width [44]:

$$D_{equilibrium} = \frac{2\gamma}{\alpha \sum P_{Zener}},$$  (13)

where $\sum P_{Zener}$ is the linear sum Zener retarding force from the grain boundary $M_{23}C_6$ carbides and Laves phase located along the LABs as well as randomly distributed MX carbonitrides. The results are summarized in Table 4.

**Table 4.** The comparison of the equilibrium lath width estimated by Equation (13) and experimental values.

| Header Conditions of the Creep Tests | Equilibrium Lath Width, µm | Equilibrium Lath Width Taking into Account Grain Boundary Particles with Sizes < 100 nm Only, µm | Experimental Lath Width, µm |
|---|---|---|---|
| Tempered | 0.49 | 0.49 | 0.29 ± 0.03 |
| 1001 h, 1% | 0.36 | 0.44 | 0.44 ± 0.03 |
| 5035 h, 2% | 0.57 | 0.76 | 0.66 ± 0.03 |
| 10,001 h, 2.2% | 0.55 | 0.73 | 0.93 ± 0.03 |
| 13,860 h, 8.4% | 0.73 | 1.11 | 0.95 ± 0.03 |

Using Equation (13), the values of the equilibrium lath width are significantly lower than the experimental lath width (Table 4). We suggest that only grain boundary particles (the $M_{23}C_6$ carbides and Laves phase) with sizes < 100 nm can cause effective Zener retarding forces, whereas the large particles do not interact with the lath boundaries under the applied stress. Modification of Equation (11), taking into account the frequency of the

particles with sizes < 100 nm from Figure 10, provides similar values of the equilibrium and experimental lath width (Table 4). So, the growth of the martensitic laths is restricted by the fine $M_{23}C_6$ carbide and Laves phase particles with sizes < 100 nm located along the LABs (Figure 13a).

In a previous study [47], 0.12 MPa is mentioned to be a critical value of the sum of the Zener retarding forces when the TMLS is retained. From 0.12 to 0.08 MPa of Zener retarding force, the TMLS starts to transform into a subgrain structure. And at Zener retarding force < 0.05 MPa, a 100% polygonal subgrain structure occurs. The present results are in agreement with the previous results [47]. In the Re-containing 10% Cr–3%Co–3% W steel, after the primary creep stage, the value of the sum of Zener forces is about 0.17 MPa that indicates retaining of the TMLS (Figure 5, Tables 2 and 4). The Laves phase plays an important role, providing about 30% of Zener retarding force (Figure 13a). During the steady-state creep stage, with a decrease in the sum of Zener forces to 0.1 MPa, the TMLS → subgrain structure transformation starts and the lath width is remarkably increased (Figure 5, Table 2) that is found to be a typical feature of low applied stresses [36]. On the other hand, for the high applied stress region (before the creep strength breakdown), the significant growth of the lath width is not revealed for the studied steel as it is for other 9–12% Cr martensitic steels [32,46,47]. The evolution of the Laves phase located along the LABs is found to be a key degradation process, which facilitates the lath growth during the steady-state creep stage for the studied steel. Both the increase in size to $160 \pm 10$ nm and dissolution of the fine particles (only 25% of the Laves phase particles have sizes < 100 nm) along the LABs (Figures 9 and 10, Table 2) eliminate the effectiveness of the Laves phase as the strengthening phase from the view of the interaction of the particles with the lath boundaries (Figure 13a). Only $M_{23}C_6$ carbides prevent the migration of the lath boundaries during the steady-state creep stage.

During the tertiary creep stage and after rupture, the lath width reaches about 1 μm (Figure 5, Tables 2 and 4) as the sum of the Zener forces reduces to 0.07 MPa (Figure 13a). This indicates that the TMLS → subgrain structure transformation continues, but formation of a 100% subgrain structure is not attained (Figure 5). Notably, ~90% of Zener force is caused by $M_{23}C_6$ carbides. Both Laves phase and MX carbonitrides slightly interact with the lath boundaries. Moreover, to estimate the lath/subgrain size via loading as $d_{load} = 10Gb/\sigma$, where σ is an applied stress, $d_{load}$ should be 1.74 μm. This indicates that Zener force due to $M_{23}C_6$ carbides prevents the significant growth of the martensitic laths under the applied stress.

- *Detachment stress of dislocation from an attractive particle after finishing the climb*

The Orowan model, which we used for estimation of contributions of particle strengthening to the room YS, is not suitable for analysis of the interaction between the secondary phase particles and free dislocations during creep at elevated temperatures. In [30,31,37], the detachment of the dislocation from an attractive particle after finishing the climb is considered to be the most plausible variant of the interaction between dislocations and particles located along the LABs under loading at elevated temperatures. Shear detachment stress ($\tau_d$) can be written as follows [48]:

$$\tau_d = Gb\sqrt{1 - K^2}/\lambda, \tag{14}$$

where *G* is the shear modulus, *b* is Burger's vector, *K* is a relaxation parameter (0.85) [48], and λ is the distance between particles located along the LABs, which is estimated as [44]:

$$\lambda = 0.5\left(\frac{2\beta}{D}\right)^{-1/2}, \tag{15}$$

where β is the number particle density (in $\mu m^{-1}$), *D* is the lath width (in μm).

Figure 13b shows change in detachment stresses from the $M_{23}C_6$ and Laves phase located along the LABs, as well as the NbX and VX carbonitrides randomly distributed

within the ferritic matrix during creep. The root mean square (rms) sum detachment stress is evaluated as $\sqrt{M23C6^2 + Laves^2 + NbX^2 + VX^2}$ as suggested in [30,31].

The detachment stresses from the $M_{23}C_6$ and NbX in the tempered state are 55 and 21 MPa, respectively; the rms sum detachment stress is 58 MPa (Figure 13b). At the end of the primary creep stage, the rms sum detachment stress remains at a high level compared with the tempered state. Although the contribution due to the $M_{23}C_6$ carbides is decreased by 22%, the detachment stress due to the Laves phase compensates this loss (Figure 13b). During the steady-state creep stage, the remarkable reduction in the detachment stresses from the boundary particles is revealed that decreases the rms sum detachment stress by 27%, although the additional contribution due to the VX phase occurs (Figure 13b). During the tertiary creep stage up to rupture, the contributions due to the $M_{23}C_6$ carbides and Laves phase to the detachment stress are slightly decreased that reduces the rms sum detachment stress by ~20%. Notably, at the tertiary creep stage (10,001 h), the contributions due to the $M_{23}C_6$ carbides located along the LABs and the NbX carbonitrides become nearly the same.

The comparison of the reduction in dislocation density within the lath interiors (Figure 8, Table 2) with the detachment stresses (Figure 13b) demonstrates that the high rms sum detachment stress of about 56 MPa provides high dislocation density within the lath interiors ($10^{14}$ m$^{-2}$).This level of rms sum detachment stress is in accordance with that in Re-free 10% Cr steel at the steady-state creep stage [30]. The retained high rms sum detachment stress overcomes the appearance of the creep strength breakdown at 923 K [27,30]. The critical value of the rms sum detachment stress of about 40 MPa at the end of the steady-state creep stage corresponds to low dislocation density within the lath interiors (~$10^{13}$ m$^{-2}$) (Figure 13b) that causes the appearance of the creep strength breakdown using the threshold stress concept [31]. This corresponds to about 30 MPa from $M_{23}C_6$ carbides, 17 MPa from the Laves phase, 24 MPa for the NbX, and 7 MPa from the VX (Figure 13b).

Therefore, the degradation in the dispersions of both Laves phase and $M_{23}C_6$ carbides located along the LABs causes the migration of lath boundaries and rearrangement/movement of free dislocations that provides the appearance of the creep strength breakdown under low applied stresses.

## 5. Conclusions

The microstructural evolution of the Re-containing 10% Cr–3% Co–3% W steel during creep at a temperature of 923 K and under the applied stress of 120 MPa was investigated. This creep condition corresponded to the low applied stress region before the appearance of the creep strength breakdown. The main results can be summarized as follows:

1.  The TMLS in the tempered state is characterized by the small width of the martensitic laths of $290 \pm 30$ nm and high dislocation density within the lath interiors of $(2.0 \pm 0.5) \times 10^{14}$ m$^{-2}$ and inside the lath boundaries of $(1.5 \pm 0.1) \times 10^{14}$ m$^{-2}$. The TMLS is stabilized by the $M_{23}C_6$ carbides with a mean size of $67 \pm 10$ nm, NbX carbonitrides with a mean size of $37 \pm 10$ nm, and $M_6C$ carbides with a mean size of $28 \pm 5$ nm.

2.  During the primary creep stage, strengthening of the TMLS is caused by the fine Laves phase particles with a mean size of $110 \pm 10$ nm that compensate the decrease in solid solution strengthening due to the decrease in W content in the ferrite. These fine Laves phase particles provide a high Zener retarding force of 0.025 MPa in the interactions with the LABs of the martensitic laths and detachment stress of 26 MPa in the interactions with free dislocations. Here, the Laves phase acts as the effective strengthening phase together with $M_{23}C_6$ carbides.

3.  During the steady-state creep stage, softening of the TMLS due to the reduction in dislocation density within the lath interiors and degradation in the dispersion of the grain boundary particles located along the LABs is observed; the TMLS is transformed into a subgrain structure. The Laves phase loses its effectiveness as the strengthening phase: both Zener force and detachment stress are dramatically

decreased. Retaining of the TMLS is provided by $M_{23}C_6$ carbides only: Zener force is 0.08 MPa and detachment stress is 30 MPa. The precipitation of the VX phase with the mean size of 55 nm does not stabilize the TMLS due to the negligible volume fraction. During the tertiary creep stage up to rupture, the mutual growth of martensitic laths and grain boundary particles located along the LABs leads to softening of the TMLS by 11%.

4. The rms sum detachment stress of 40 MPa is the critical value, below which low dislocation density within the lath interiors occurs. Both low dislocation density within the lath interior and the decrease in the detachment stresses from the Laves phase and $M_{23}C_6$ carbides are considered to be the origin of the appearance of the creep strength breakdown.

**Author Contributions:** Conceptualization, A.F.; methodology, A.F.; formal analysis, I.B., S.D. and I.N.; investigation, A.F. and I.N.; writing—original draft preparation, A.F.; writing—review and editing, A.F.; visualization, A.F.; supervision, A.F. and R.K.; funding acquisition, A.F. All authors have read and agreed to the published version of the manuscript.

**Funding:** This research was funded by Russian Science Foundation, grant number 19-73-10089-П. https://rscf.ru/project/19-73-10089/ (accessed on 22 August 2023).

**Data Availability Statement:** The data presented in this study are available on request from the corresponding author. The data are not publicly available because the data also form part of an ongoing study.

**Acknowledgments:** The work was carried out using equipment of the Joint Research Center of Belgorod State National Research University "Technology and Materials".

**Conflicts of Interest:** The authors declare no conflict of interest.

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
