# Peer review of "Microstructural Evolution of a Re-Containing 10% Cr-3Co-3W Steel during Creep at Elevated Temperature"

_metals, doi:10.3390/met13101683_

Round 1
Reviewer 1 Report
Review comment to the manuscript metals-2599426
Manuscript No.: metals-2599426
Title: Microstructural evolution of a Re-containing 10% Cr-3Co-3W-2 steel during creep at elevated temperature
Authors: Alexandra Fedoseeva, Ivan Brazhnikov, Svetlana Degtyareva, Ivan Nikitin and Rustam Kaibyshev
General comments
10% low-C, high-B, Cr-rich martensitic steels containing low C was investigated in this work with emphasis of creep properties affected by strengthening mechanisms, carbides and Laves phase. The microstructure of the steels and their mechanical properties were investigated in detail which supported the conclusion about the reasons of steel softening. Long-time creep experiments is very interested and impressing. A couple of microscopic methods were applied to reveal microstructure of the prepared steels and their time-of-fly creep properties. This work is well written and enriched with nice mechanical properties measurement. It merits to the scope for publishing at MDPI Metals.
Special comments
The overall experimental section has been in detailed description. The investigation of the microstructure and the creep properties are impressing with TEM and EBSD techniques. Besides the convincing data analysis, just a few minor points may draw the authors’ attention:
1. There is in lack of a summary of the phases investigated in this work. The crystallographic structure data of quite a few phases, including M23C6, Laves, M6C, NbX and VX, is suggested to show in a table covering space group, Bravias lattice type and lattice parameters. Such crystallographic data is also missing in the cited paper (ref. [28]) although it is expected to find there. With no such information, it is difficult to judge if it is correct in phase identification, especially electron diffraction pattern indexing.
2. Both in the ref. [28] and this work, there is no chemical analysis data support to identify phases. It is expected to see EDS point scan, line profile or elemental mapping as evidence of chemical distribution for phase identification if available or possible.
3. As shown in Figure 1b and 1c, the M23C6 and NbX phases are in a size that is suitable for micro-electron beam or nanobeam electron diffraction to give an electron diffraction pattern from a single particle only. This will be a benefit of phase identification by using zone-axis diffraction pattern.
4. The name of the phase NbX should be denoted with the meaning of the letter “X”. Is it for Carbon, Nitrogen, or both?
5. Figure 2: Although the pseudo-coloured EBSD figures provide straightforward overview of grain boundary angle diversity, it would be good to identify the multiplicity number of the corresponding coincidence site lattice (CSL) for the observed grain boundaries. The EBSD software should have such function for auto calculation of CSL.
6. When estimating volume fraction of the particles, the way to calculate number particle density is not described. Did the authors count manually, or a software tool was used for imaging analysis? What is the error tolerance of volume fraction?
7. When estimating dislocation density in martensitic laths, the Burger’s vector of the observed dislocations in martensitic laths was not listed in Table 1. The question is what it is and how it is determined or measured experimentally. It is apparent that smaller Burger’s vector gives higher dislocation density. Please explain this for TEM method and SEM method sperately.
8. Table 1 gives very different dislocation density provided by TEM and EBSD separately. However, the main text claims that they are similar. Please specify if they are different or similar and which technique is more reliable.
9. EDP in Figure 4 shows a bright spot corresponding to transmitted beam. However, it located away from the diffraction pattern centre. Is it plot by the authors or is it due to two-exposure method?
Conclusion
This is a very good paper on metallic engineering materials in terms of the relationship between mechanical properties and the microstructure evolution during mechanical testing. Detailed experimental data and the data analysis provides abundant evidence to support the conclusion. The long-time creep experiment and its large dataset is very useful for other researchers’ reference. It is close to the status of acceptance of publishing. But it is hopeful to give the authors a chance for their consideration of the above minor comments.
Author Response
Point 1. There is in lack of a summary of the phases investigated in this work. The crystallographic structure data of quite a few phases, including M23C6, Laves, M6C, NbX and VX, is suggested to show in a table covering space group, Bravias lattice type and lattice parameters. Such crystallographic data is also missing in the cited paper (ref. [28]) although it is expected to find there. With no such information, it is difficult to judge if it is correct in phase identification, especially electron diffraction pattern indexing.
Response 1: Thank you for fruitfull suggestion. All information about phases investigated in work was summarized in Table 1 (p. 4).
Table 1. A summary of the phases investigated in the work.
|
Phase |
Space group |
Bravias lattice type |
Lattice parameters, nm |
|
M23C6 |
Fmm |
face-centered cubic |
a=1.0656 |
|
NbC |
Fmm |
face-centered cubic |
a=0.44698 |
|
M6C (Fe3W3C) |
Fdm |
face-centered cubic |
a=1.1087 |
|
Laves (Fe2W) |
P6(3)/mmc |
hexagonal |
a=0.4727; c=0.7704 |
|
VN |
Fmm |
face-centered cubic |
a=0.413916 |
Point 2: Both in the ref. [28] and this work, there is no chemical analysis data support to identify phases. It is expected to see EDS point scan, line profile or elemental mapping as evidence of chemical distribution for phase identification if available or possible.
Response 2: EDS point scans for M23C6, NbX, Laves phase, and VX were added to Figures 1b, 1c, 5, and 11b.
Point 3: As shown in Figure 1b and 1c, the M23C6 and NbX phases are in a size that is suitable for micro-electron beam or nanobeam electron diffraction to give an electron diffraction pattern from a single particle only. This will be a benefit of phase identification by using zone-axis diffraction pattern.
Response 3: Thank your for your comment. The particles lie on the surface of foil, and EDP from a particle contains the reflexes from both particle and ferritic matrix. The mean size of M23C6 carbides of 40 nm and NbX of 30 nm allows using micro-electron beam or nanobeam electron diffraction to obtain EDP from a one single particle only. However, the mutual EDP from particle and matrix contain usefull information about orientation relationship between matrix and particles. Moreover, we used the carbon replicas to obtain the clean EDP from the particles without an effect of ferritic matrix.
Point 4: The name of the phase NbX should be denoted with the meaning of the letter “X”. Is it for Carbon, Nitrogen, or both?
Response 4: Thank your for your comment. The text was added to Manuscript: “The stability of TMLS in the initial state is reached by the precipitation of M23C6 carbides along all HABs and LABs of TMLS, as well as MX carbonitrides (where M means V and/or Nb or their combination, X means C and/or N or their combination) randomly distributed in ferritic matrix [5-10].” (p. 1).
Point 5: Figure 2: Although the pseudo-coloured EBSD figures provide straightforward overview of grain boundary angle diversity, it would be good to identify the multiplicity number of the corresponding coincidence site lattice (CSL) for the observed grain boundaries. The EBSD software should have such function for auto calculation of CSL.
Response 5: Thank you for your usefull suggestion. We tried to determine the multiplicity number of the corresponding coincidence site lattice (CSL), and Σ3 is found to be the dominant CSL boundaries. These Σ3 boundaries can correspond to twin boundaries; on the other hand, 60° boundary is one of variants of Kurdjumov–Sachs ORs for fcc→bcc transformation. We suggest that the addition of Σ3 boundaries to EBSD figure can confuse the readers.
Point 6: When estimating volume fraction of the particles, the way to calculate number particle density is not described. Did the authors count manually, or a software tool was used for imaging analysis? What is the error tolerance of volume fraction?
Response 6: The authors count the particle number density manually. The methodic of estimation of particle number density was added to Manuscript: “The particle number density was estimated as count of the particles per unit of lath length projection [27]. (p. 4)”. The volume fraction of particles located along LABs in Table 1 is calculated using the particle number density, particle size and lath width, and error of the volume fraction is represented in Table 1.
Point 7: When estimating dislocation density in martensitic laths, the Burger’s vector of the observed dislocations in martensitic laths was not listed in Table 1. The question is what it is and how it is determined or measured experimentally. It is apparent that smaller Burger’s vector gives higher dislocation density. Please explain this for TEM method and SEM method sperately.
Response 7: We don’t determine the Burger’s vector neigther TEM or SEM. We suggested the Burger vector of 2.48 × 10-10 m assuming all of the dislocations in the steel have a Burger vector of the type 1/2(111).
Point 8: Table 1 gives very different dislocation density provided by TEM and EBSD separately. However, the main text claims that they are similar. Please specify if they are different or similar and which technique is more reliable.
Response 8: Actually, Table 1 contains dislocation densities estimated by TEM (this is dislocation density within martensitic laths), and EBSD using (i) lath area per unit volume and average lath misorientation (it is dislocation density inside lath boundaries) and (ii) KAM value. After tempering, the values of dislocation densities estimated by TEM and EBSD via lath area per unit volume and average lath misorientation was similar ((2.0±0.5)×1014 and (1.5±0.1)×1014 m-2), whereas dislocation density estimated via KAM value has higher value that is caused by the fact that KAM included the geometrically necessary dislocations and low-angle lath boundaries with misorientations <5 deg. After creep, the values of dislocation density estimated via TEM and EBSD significantly differ that is related to limit of EBSD analysis. In the article, we concluded that EBSD analysis is poorly described the materials with well-developed system of the low-angle boundaries with θ <2 deg (p. 12).
Point 9: EDP in Figure 4 shows a bright spot corresponding to transmitted beam. However, it located away from the diffraction pattern centre. Is it plot by the authors or is it due to two-exposure method?
Response 9: Thank you for correction. Sure, bright spot corresponding to transmitted beam is located in the diffraction pattern centre. Authors cut the part of diffraction pattern to indicate the Laves phase and to arrange all captions on EDP. Althouth, we actually used two-exposure method.

Reviewer 2 Report
Review of the manuscript 2599426The title is “Microstructural evolution of a Re-containing 10% Cr-3Co-3W steel during creep at elevated temperature.” Authors observed microstructures details of 10% Cr steel. I think these data seem to be correct, but I have some comments as follows.
1. Page 2, line 71, Autor use only 10% Cr, and same pre-heat treatment (homogenization, annealing), so the initial volume fraction and shape of precipitations like M23C6, Laves, M6C are same. I think that scientific interest are effect of Cr content and precipitations volume and shape on the properties of this steel (Cr-Co-W-Fe).
2. Page 3, line 118-120, The volume fraction of M23C6 is 2 % same in this experiment from beginning to end of creep. I cannot judge that existence of M23C6 effects on the creep properties or not. I think that the same goes for Laves.
3. Page 4, line 146, Authors observed the value of KAM by TEM. I am interested about KAM, but the value is same in this experiment. Does KAM change depending on conditions, and does it affect creep behavior?
4. Figure 9, Page 11, line 309, Why the particles coarsening occurred along HABs, mainly?
5. Page 14-15, line 392-406, Authors mentioned about Solid solution strengthening, Particle strengthening and Lath/subgrain strengthening. These are the three basic strengthening mechanisms. However, to discuss these, it is necessary to conduct experiments with samples with different Cr content, different M23C6 and different Lath/subgrain size. In my experiment, these parameters need to change at least three steps. In my experience, those data need to be varied in at least three stages.
These are my comments.
Author Response
Point 1. Page 2, line 71, Autor use only 10% Cr, and same pre-heat treatment (homogenization, annealing), so the initial volume fraction and shape of precipitations like M23C6, Laves, M6C are same. I think that scientific interest are effect of Cr content and precipitations volume and shape on the properties of this steel (Cr-Co-W-Fe).
Response 1: Thank you for your comment. No doubts that effect of Cr content is very interesting for analysis of mechanical properties for Cr-Co-W-Fe base. On the other hand, the evolution of tempered martensite lath structure is revealed during creep test and leads to the creep strength breakdown appearance. Understanding the regularities of structural degradation for 10% Cr steel together with the comparison with other 9-12% Cr steels can help us to develop the common theory for creep strength of these steels. I hope that we will write the review collecting all our knowledges about the structures and mechanical properties of 9-12% Cr steels in the near future.
Point 2: Page 3, line 118-120, The volume fraction of M23C6 is 2 % same in this experiment from beginning to end of creep. I cannot judge that existence of M23C6 effects on the creep properties or not. I think that the same goes for Laves.
Response 2: Actually, the volume fraction of M23C6 of 2% was unchangeable during creep test up to rupture. This was characterized for Laves phase too. On the other hand, the size and particle number density of these phases located along both LABs and HABs were changed during creep; and this evolution determines the creep behavior of this steel.
Point 3: Page 4, line 146, Authors observed the value of KAM by TEM. I am interested about KAM, but the value is same in this experiment. Does KAM change depending on conditions, and does it affect creep behavior?
Response 3: The value of KAM was observed by EBSD analysis. KAM in the tempered state and in the end of creep test had the same value. We didn’t find the correlation between KAM and creep behavior.
Point 4: Figure 9, Page 11, line 309, Why the particles coarsening occurred along HABs, mainly?
Response 4: We suggested that coarsening of M23C6 carbides occurred along HABs, mainly, since the increase in the size of these particles was more intensive along HABs, and higher fraction of large particles was appeared along HABs.
Point 5: Page 14-15, line 392-406, Authors mentioned about Solid solution strengthening, Particle strengthening and Lath/subgrain strengthening. These are the three basic strengthening mechanisms. However, to discuss these, it is necessary to conduct experiments with samples with different Cr content, different M23C6 and different Lath/subgrain size. In my experiment, these parameters need to change at least three steps. In my experience, those data need to be varied in at least three stages.
Response 5: At analysis of strengthening mechanisms, we calculated the contributions of all strengthening mechanisms: dislocation strengtheing, solid solution strengthening, particle strengthening, and lath/subgrain strengthening. All these contributions had the changes during creep tests due to the evolution of structure: dislocation density decreased, W and Mo left the solid solution; M23C6 carbides and Laves phase coarsened, the lath width increased with formation of new subgrains. We tried to take into account all changes in the structure during creep and to show how it affected the creep behavior.